# β-catenin and γ-catenin are dispensable for T lymphocytes and AML leukemic stem cells

Xin Zhao[1†], Peng Shao[2†], Kexin Gai[1], Fengyin Li[3], Qiang Shan[1], Hai-Hui Xue[1,4*]

[1]Center for Discovery and Innovation, Hackensack University Medical Center, Nutley, United States; [2]Department of Microbiology and Immunology, Carver College of Medicine, University of Iowa, Iowa City, United States; [3]Hefei National Laboratory for Physical Sciences at Microscale, the CAS Key Laboratory of Innate Immunity and Chronic Disease, School of Basic Medical Sciences, Division of Life Sciences and Medicine, University of Science and Technology of China, Hefei, China; [4]New Jersey Veterans Affairs Health Care System, East Orange, United States

**Abstract** The β-catenin transcriptional coregulator is involved in various biological and pathological processes; however, its requirements in hematopoietic cells remain controversial. We re-targeted the *Ctnnb1* gene locus to generate a true β-catenin-null mutant mouse strain. Ablation of β-catenin alone, or in combination with its homologue γ-catenin, did not affect thymocyte maturation, survival or proliferation. Deficiency in β/γ-catenin did not detectably affect differentiation of CD4[+]T follicular helper cells or that of effector and memory CD8[+] cytotoxic cells in response to acute viral infection. In an MLL-AF9 AML mouse model, genetic deletion of β-catenin, or even all four Tcf/Lef family transcription factors that interact with β-catenin, did not affect AML onset in primary recipients, or the ability of leukemic stem cells (LSCs) in propagating AML in secondary recipients. Our data thus clarify on a long-standing controversy and indicate that β-catenin is dispensable for T cells and AML LSCs.

**\*For correspondence:** haihui.xue@hmh-cdi.org

[†]These authors contributed equally to this work

**Competing interests:** The authors declare that no competing interests exist.

## Introduction

β-catenin is a known transcriptional coregulator that interacts with transcription factors in the Tcf/Lef family and others to modulate gene expression (*Cadigan, 2012*; *Mosimann et al., 2009*). β-catenin protein is regulated through post-translational modifications. Phosphorylation at a cluster of Ser/Thr residues in its N-terminus results in proteasome-mediated degradation of β-catenin (*Cadigan, 2012*; *Mosimann et al., 2009*). Activation of several signaling pathways such as Wnt and prostaglandin E2 leads to inactivation of the kinases that are responsible for β-catenin phosphorylation and therefore accumulation of β-catenin protein (*Goessling et al., 2011*; *Klaus and Birchmeier, 2008*). Due to its strong impact on transcription, the activity of β-catenin is tightly controlled, with aberrant β-catenin activation frequently associated with malignant transformation and various cancers (*Clevers and Nusse, 2012*). In the hematopoietic system, modest activation of β-catenin has been shown to have beneficial effects, such as extending survival of thymocytes (*Xie et al., 2005*) and regulatory T cells (*Ding et al., 2008*), promoting expansion of memory CD8[+] T cells (*Gattinoni et al., 2009*; *Zhao et al., 2010*). However, excessive β-catenin activation, through deletion of exon 3 of *Ctnnb1* gene (which encodes the Ser/Thr cluster in β-catenin protein), has detrimental effects on the function of hematopoietic stem cells (HSCs) (*Kirstetter et al., 2006*; *Scheller et al., 2006*), blocks thymocyte maturation and promotes thymocyte transformation (*Guo et al., 2007b*).

Whereas it is clear that β-catenin activation bears strong biological effects on blood cells, the requirement for β-catenin has been controversial. During thymocyte maturation, for example, genetic deletion of exons 3–6 of the *Ctnnb1* gene caused modest developmental blocks and modest reduction in thymic cellularity (*Xu et al., 2003*). In other reports, however, no thymocyte maturation defects were observed when *Ctnnb1* exons 2–6 were inducibly deleted with Mx1-Cre (*Cobas et al., 2004*), or in chimeric mice reconstituted with fetal liver cells lacking β-catenin and its homologue, γ-catenin (*Jeannet et al., 2008*; *Koch et al., 2008*). Additionally, mature CD8+ T cells in these β-catenin-targeted models showed intact response to viral infections (*Driessens et al., 2010*; *Prlic and Bevan, 2011*). On the other hand, among the Tcf/Lef family transcription factors (TFs) that interact with β-catenin, Tcf1 and Lef1 are expressed in T lineage cells (*Staal et al., 2008*; *Xue and Zhao, 2012*). Null mutations of Tcf1 alone or together with Lef1 show more profound T cell developmental blocks and more severe decrease in thymic cellularity (*Germar et al., 2011*; *Okamura et al., 1998*; *Verbeek et al., 1995*; *Weber et al., 2011*; *Yu et al., 2012b*). Recent studies also revealed multifaceted roles of Tcf1 in mature T cell responses including differentiation of follicular helper T cells (*Choi et al., 2015*; *Raghu et al., 2019*; *Wu et al., 2015*; *Xu et al., 2015*). These discrepancies have posed a major challenge in the past two decades as to the true requirements for β-catenin and its connection with Tcf/Lef TFs in hematopoietic cells.

One notable observation is that both *Ctnnb1*-targeted strains used in previous studies retain a truncated protein in hematopoietic cells (*Jeannet et al., 2008*). The *Ctnnb1* gene has 15 exons, deletion of exons 2–6 or exons 3–6 in both models (*Brault et al., 2001*; *Huelsken et al., 2000*) may have allowed in-frame translation from downstream exons, giving rise to an N-terminally truncated β-catenin protein of 40–50 kDa. Because the N-terminus of β-catenin contains phosphorylation sites for ubiquitin-dependent degradation, an N-terminally truncated form of β-catenin protein has longer half-life, and its ectopic expression has been shown to stimulate proliferation and apoptosis of intestinal crypts (*Wong et al., 1998*). In addition, a C-terminally truncated β-catenin is a naturally occurring β-catenin paralog in planarians, and acts as a negative regulator of full-length β-catenin during planarian eye photoreceptor specification (*Su et al., 2017*). Therefore, the presence of the truncated β-catenin protein in previously β-catenin-targeted models may functionally compensate for loss of full-length β-catenin, underlying the lack of severe defects in T cell development or mature T cell responses. On the other hand, the truncated β-catenin protein may function as a dominant negative, and this possibility then raises questions on observed requirements for β-catenin using the existing β-catenin-targeted models. For example, β-catenin is considered essential for self-renewal of leukemic stem cells (LSCs) in both chronic and acute myeloid leukemia (CML and AML, respectively) (*Hu et al., 2009*; *Wang et al., 2010*; *Yeung et al., 2010*; *Zhao et al., 2007*). This work aims to address these long-standing questions.

## Results and discussion

We previously obtained the *Ctnnb1* exons 2–6 floxed model (*Brault et al., 2001*). We aimed to ablate β-catenin protein in all hematopoietic cells using Vav-Cre, but found accumulation of a truncated form of β-catenin protein of approximately 40 kDa in bone marrow (BM) cells (*Figure 1—figure supplement 1*), consistent with a previous report (*Jeannet et al., 2008*). To achieve complete ablation of β-catenin protein, we re-targeted the *Ctnnb1* gene locus by inserting LoxP sites into introns 1 and 14, respectively (*Figure 1A*) so that a Cre recombinase removes 13 of a total of 15 *Ctnnb1* exons. The resulting *Ctnnb1*-floxed mice were crossed with a strain that expresses Cre recombinase and estrogen receptor fusion protein (Cre-ERT2) in the ubiquitously expressed *Rosa26* locus (*Guo et al., 2007a*). We maintained a single *Rosa26*Cre-ERT2 allele in all mice used in this study (referred to as CreER+ for simplicity), so as to avoid dose-dependent effects by tamoxifen and/or Cre. Lineage-negative BM cells and peripheral T cells from CreER+*Ctnnb1*fl/fl mice were cultured in vitro with tamoxifen or its carrier solvent, DMSO, and immunoblotting validated effective ablation of β-catenin protein without generating truncated form(s) (*Figure 1B*). A band of 50 kDa was detected in both tamoxifen- and DMSO-treated T cells, which was considered non-specific because its size was larger than the truncated β-catenin protein resulting from translation from downstream exons (compare with *Figure 1—figure supplement 1*). These analyses validate that the new targeting strategy can generate a true β-catenin null mutation.

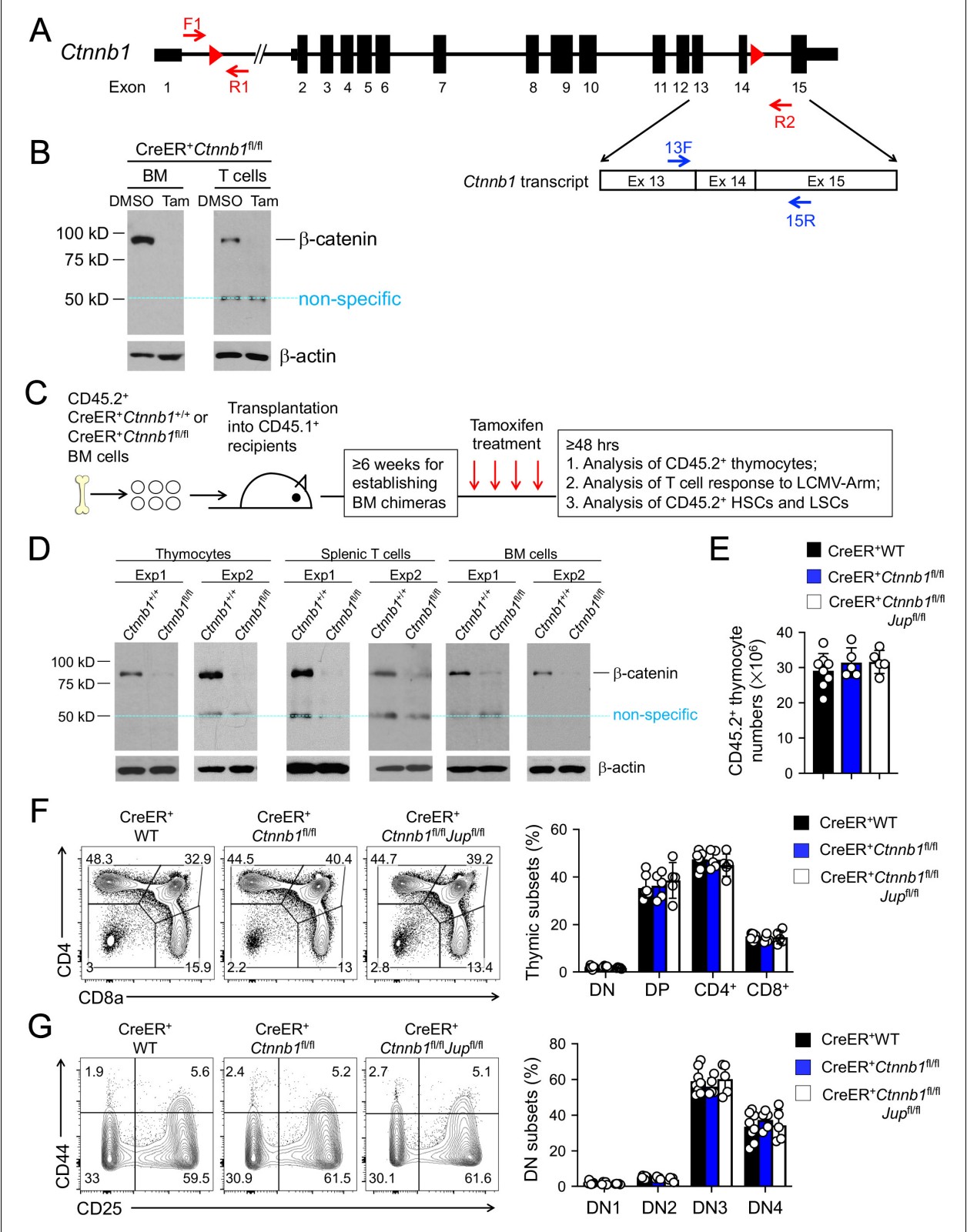

**Figure 1.** β-catenin null mutation alone or combined deletion with γ-catenin did not detectably affect thymocyte development. (**A**) Targeting strategy to generate β-catenin null mutant mouse strain. All *Ctnnb1* exons are shown, with red triangles denoting LoxP sites and red arrows denoting genotyping primers. Shown in the lower panel is partial *Ctnnb1* transcript with blue arrows marking RT-PCR primers. (**B**) Ex vivo β-catenin ablation. Lineage-negative BM cells and splenic CD3[+] T cells were isolated from CreER[+]*Ctnnb1*[fl/fl] mice and cultured with DMSO or tamoxifen followed by

*Figure 1 continued on next page*

*Figure 1 continued*

immunoblotting for β-catenin and β-actin with the latter as equal loading control. (C) Experimental design for generation of BM chimeras and analysis. (D) In vivo β-catenin ablation. BM chimeras reconstituted with WT or CreER$^+$Ctnnb1$^{fl/fl}$ BM cells were treated with tamoxifen as in C). CD45.2$^+$ total thymocytes, TCRβ$^+$ splenocytes, or total BM cells were sorted and immunoblotted for β-catenin and β-actin. Data from two independent experiments are shown. In (B and D), the 50 kDa band that appeared in some blots are considered non-specific reactivity to the anti-β-catenin antibody. Refer to *Figure 1—figure supplement 1* for size comparison with a truncated β-catenin protein produced from *Ctnnb1* exons 2–6-targeted allele. (E) Thymic cellularity. WT, CreER$^+$Ctnnb1$^{fl/fl}$, or CreER$^+$Ctnnb1$^{fl/fl}$Jup$^{fl/fl}$ BM chimeras were treated with tamoxifen as in C), and CD45.2$^+$ thymocytes were enumerated. (F) Detection of thymic maturation stages. CD45.2$^+$ thymocytes were surface-stained with biotinylated lineage markers (minus CD3ε) to exclude non-T cells, and with CD4 and CD8 to identify DN, DP, CD4$^+$ and CD8$^+$ subsets. (G) Detection of DN subsets. CD45.2$^+$ DN thymocytes were surface-stained with CD44 and CD25 to identify DN1 to DN4 subsets. In panels (E–G), values in representative contour plots denote percentages, and bar graphs are cumulative data of means ± s.d. from ≥3 experiments. None of the parameters was statistically significant among the groups as determined by one-way ANOVA, and thus unmarked for clarity.

The online version of this article includes the following source data and figure supplement(s) for figure 1:

**Source data 1.** Source files, containing original data for *Figure 1E,F and G*, to document thymic cellularity (E), frequency of thymocyte subsets at different developmental stages (F, G).

**Figure supplement 1.** Deletion of *Ctnnb1* exons 2–6 gives rise to a truncated β-catenin protein in bone marrow cells.

**Figure supplement 2.** Induced deletion of β-catenin in whole body results in mouse lethality.

**Figure supplement 3.** Validation of β-catenin null mutation on genomic DNA (A) and transcript (B) levels upon treatment of the BM chimera with Tamoxifen.

**Figure supplement 4.** Validation of γ-catenin ablation induced by tamoxifen treatment.

**Figure supplement 5.** Induced deletion of β-catenin or both β- and γ-catenin did not detectably affect thymocytes proliferation (A) or apoptosis (B).

When the whole mice were treated with tamoxifen in vivo for four consecutive days, however, CreER$^+$Ctnnb1$^{fl/fl}$ mice exhibited morbidity within 10 days, while CreER$^+$Ctnnb1$^{+/+}$ mice remained healthy (*Figure 1—figure supplement 2*), indicating ablating β-catenin in multiple organs may have severely compromised vital functions. To specifically address the function of β-catenin in hematopoietic cells, we transplanted BM cells from CD45.2$^+$ CreER$^+$Ctnnb1$^{+/+}$ (WT) or CreER$^+$Ctnnb1$^{fl/fl}$ mice into irradiated CD45.1$^+$ recipients (*Figure 1C*). After ≥6 weeks when the BM chimeras were stably established, treatment of the recipients with the same tamoxifen regimen did not cause lethality (not shown). Effective ablation of β-catenin was observed in sorted CD45.2$^+$CreER$^+$Ctnnb1$^{fl/fl}$ thymocytes, splenic T cells, and total BM cells (*Figure 1D*). We further determined the efficacy of deleting floxed exons in genomic DNAs, where the PCR products from F1 and R1 primers were detected from the WT *Ctnnb1* allele (*Figure 1A*), but were barely detectable in sorted CD45.2$^+$ BM cells from tamoxifen-treated CreER$^+$Ctnnb1$^{fl/fl}$ recipients (*Figure 1—figure supplement 3A*). On the other hand, PCR products from the F1 and R2 primer combination were only detected after the intervening exons were deleted (*Figure 1—figure supplement 3A*). Effective deletion of *Ctnnb1* transcripts was also validated by RT-PCR using primer 13F (complementary to the deleted exon 13) in combination with primer 15R (complementary to undeleted exon 15) (*Figure 1—figure supplement 3B*). These characterizations further validated that null mutation of β-catenin is achieved in vivo in all hematopoietic cells in the BM chimeras.

To directly address the functional redundancy between β-catenin and its homologue γ-catenin (encoded by *Jup*), we crossed the *Jup*$^{fl/fl}$ strain (*Demireva et al., 2011*) with CreER$^+$Ctnnb1$^{fl/fl}$ mice to produce CreER$^+$Ctnnb1$^{fl/fl}$Jup$^{fl/fl}$ mice and generated BM chimeras as in *Figure 1C*. Excision of the floxed *Jup* exons 3–5 after tamoxifen treatment was validated in genomic DNAs from sorted CD45.2$^+$ BM cells with similar strategy as above (*Figure 1—figure supplement 4A*). The PCR products from *Jup* F1 and R1 primers detected from the WT *Jup* allele in WT cells were greatly diminished in tamoxifen-treated CreER$^+$Ctnnb1$^{fl/fl}$Jup$^{fl/fl}$ cells, while PCR products from *Jup* F1 and R2 primer combination were only detected in the latter (*Figure 1—figure supplement 4B*). On the mRNA level, *Jup* transcripts were only one fiftieth as abundant as *Ctnnb1* transcripts in WT cells, but both *Jup* and *Ctnnb1* transcripts were effectively deleted in tamoxifen-treated CreER$^+$Ctnnb1$^{fl/fl}$Jup$^{fl/fl}$ BM cells (*Figure 1—figure supplement 4C*). Consistent with low *Jup* transcript abundance, detection of γ-catenin protein in WT or targeted BM cells proved to be challenging; however, γ-catenin protein was readily detectable in WT hepatocytes and was greatly diminished in hepatocytes in tamoxifen-treated CreER$^+$Ctnnb1$^{fl/fl}$Jup$^{fl/fl}$ mice (non-BM chimeras, *Figure 1—figure supplement 4D*). These analyses validated effective ablation of γ-catenin in our experimental system.

Following tamoxifen-induced target gene deletion in the chimeras, total thymic cellularity was similar in WT chimeras and those lacking β-catenin alone or both β-catenin and γ-catenin (*Figure 1E*). Thymocyte maturation follows sequential stages from $CD4^-CD8^-$ double negative (DN) to $CD4^+CD8^+$ double positive (DP) followed by further differentiation to $CD4^+$ or $CD8^+$ single positive cells. Each thymic subset was detected at similar frequency among all genotypes examined (*Figure 1F*). By intracellular staining of Ki67, DN and DP cells exhibited more active proliferation than $CD4^+$ and $CD8^+$ thymocytes, and each subset showed similar frequency of $Ki67^+$ cells among all genotypes (*Figure 1—figure supplement 5A*). Detection of active Caspase-3/7 revealed that DN thymocytes showed modestly increased susceptibility to apoptosis than thymocytes in later stages, and each subset showed similar frequency of $Caspase-3/7^+$ cells among all genotypes (*Figure 1—figure supplement 5B*). In addition, subfractionating the DN thymocytes based on CD25 and CD44 expression showed that distribution of DN1-4 subsets was similar among all genotypes (*Figure 1G*). A cohort of $CreER^+$ WT and $CreER^+Ctnnb1^{fl/fl}Jup^{fl/fl}$ BM chimeras (n = 5 each) was monitored for 16 weeks after tamoxifen treatment, and none of these mice showed signs of malignant transformation of thymocytes or other hematopoietic lineage cells. These analyses showed that complete loss of β-catenin or both β- and γ-catenin did not detectably cause T cell developmental blocks, or alterations in thymocyte proliferative capacity or survival.

In response to acute viral infections, antigen-specific mature $CD4^+$ and $CD8^+$ T cells mount protective immune responses by clonal expansion and differentiation into functional effector cells. To test the requirements for β-catenin and γ-catenin in mature T cell responses, we infected tamoxifen-treated BM chimera with lymphocytic choriomeningitis virus Armstrong strain (LCMV-Arm) to elicit acute viral infection. Activated $CD4^+$ T cells predominantly differentiate into $CXCR5^-SLAM^{hi}$ T helper 1 ($T_H1$) and $CXCR5^+SLAM^{lo}$T follicular helper ($T_{FH}$) lineage cells (*Crotty, 2014*). $T_{FH}$ cells depend on Tcf1 and Lef1 for Bcl6 induction and further differentiation (*Choi et al., 2015*; *Wu et al., 2015*; *Xu et al., 2015*); in contrast, deletion of β-catenin alone or in combination with γ-catenin did not affect the frequency of $T_H1$ or $T_{FH}$ cells at the peak response (*i.e.*, 8 days post-infection, *dpi*) (*Figure 2A*). In addition, Bcl6-expressing germinal center (GC)-$T_{FH}$ cells were detected at similar frequency among all genotypes (*Figure 2B*), suggesting that β-catenin and γ-catenin are not essential for $T_{FH}$ cell differentiation. Whereas $T_{FH}$ cells maintain Tcf1 expression similar to naïve $CD4^+$ T cells (*Gullicksrud et al., 2017*), fully differentiated cytotoxic effector $CD8^+$T cells downregulate Tcf1 (*Gullicksrud et al., 2017*; *Zhao et al., 2010*). Consistent with this Tcf1 expression pattern, loss of Tcf1 has little impact on differentiation of effector $CD8^+$T cells (*Shan et al., 2017*; *Zhou et al., 2010*). Not surprisingly, deletion of β-catenin alone or both β/γ-catenin did not detectably affect generation of antigen-specific effector $CD8^+$T cells, identified as IFN-γ-producing cells upon stimulation with peptides derived from the LCMV glycoprotein (GP) 33 epitope (*Figure 2C*). The GP33-specfic effector $CD8^+$T cells showed similar capacity of producing TNF-α and similar expression of granzyme B among all genotypes (*Figure 2C,D*), suggesting that β-catenin and γ-catenin are dispensable for $CD8^+$ T cells to differentiate into functional cytotoxic T cells.

After the peak responses that resolve acute infections, a fraction of antigen-specific T cells persists as memory T cells (*Martin and Badovinac, 2014*). Tcf1 is expressed in long and short isoforms with the former having the capacity of interacting with β-catenin through its N-terminal domain (*Staal et al., 2008*; *Xue and Zhao, 2012*). Previously we ablated the Tcf1 long isoform, which led to profound reduction of memory $T_H1$ and $T_{FH}$ cells (*Gullicksrud et al., 2017*). Tcf1 is also required for memory $CD8^+$ T cells (*Jeannet et al., 2010*; *Zhou et al., 2010*). To determine if β-catenin and γ-catenin are required for memory T cells, we analyzed the infected BM chimeras at ≥40 *dpi*. During 15–16 and 30–31 *dpi*, we added two rounds of tamoxifen treatment to the LCMV-infected BM chimeras, so as to prevent rebound of undeleted cells and ensure persistent deletion of β/γ-catenin proteins. $CXCR5^+$ memory $T_{FH}$ cells were detected at similar frequency in WT and β/γ-catenin-null BM chimeras (*Figure 2E*). GP33-specific memory $CD8^+$ T cells were not diminished in β/γ-catenin-null BM chimeras and preserved similar capacity of producing TNF-α (*Figure 2F*). Collectively, loss of β-catenin and γ-catenin did not detectably affect T cell responses at either effector or memory phase.

β-catenin ablation using previously *Ctnnb1*-targeted models did not show potent impact on maintenance of HSCs (*Ruiz-Herguido et al., 2012*; *Zhao et al., 2007*). Consistent with these observations, null mutation of β-catenin alone or both β/γ-catenin did not affect BM cellularity (*Figure 3A*), or the frequency of $Lin^-Sca1^+c-Kit^+$ (LSK) cells, in which HSCs were enriched (*Figure 3B*). HSCs with long-term repopulation capacity, as marked by $CD150^+CD48^-$ phenotype, were detected at similar

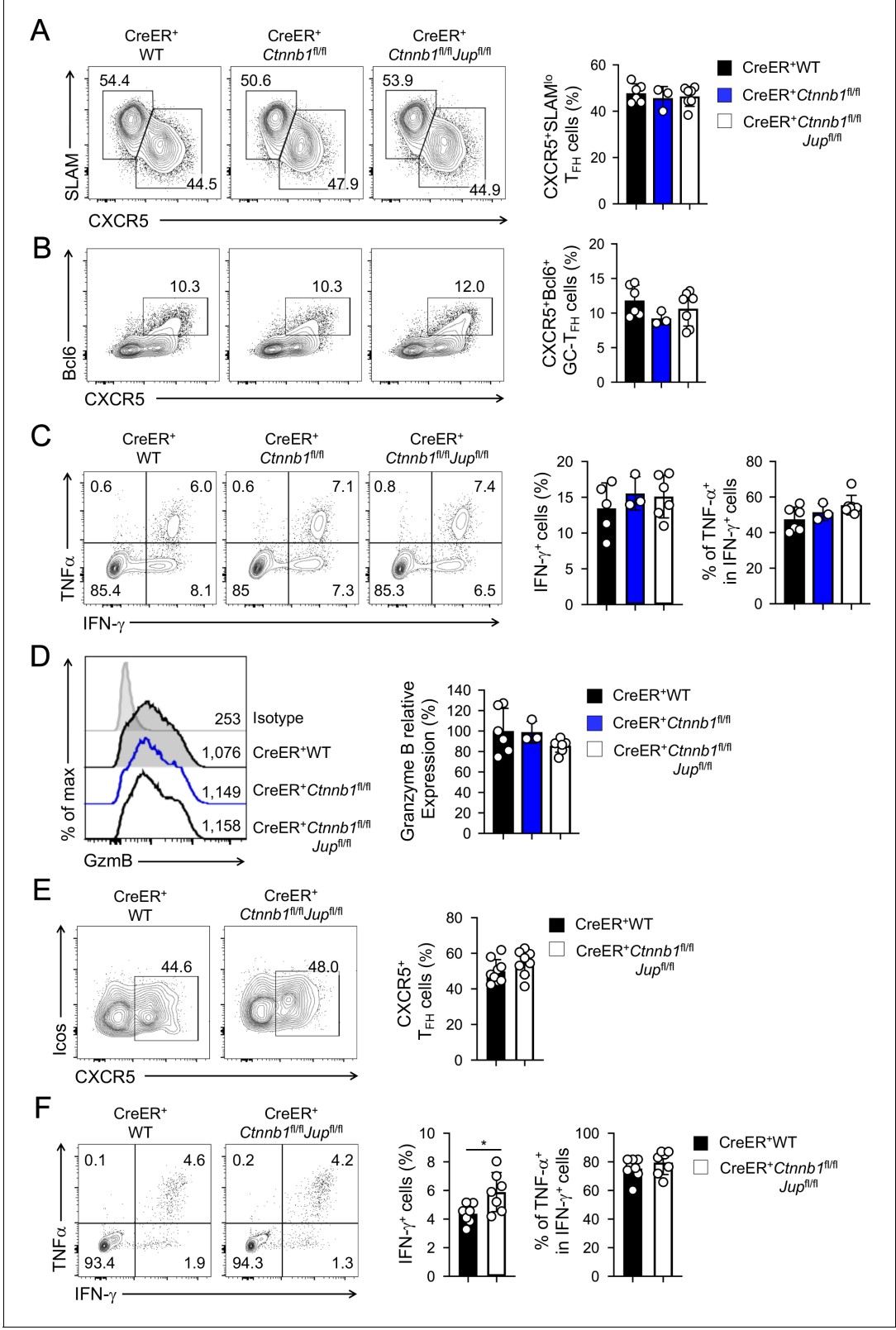

**Figure 2.** β-catenin and γ-catenin are not required for T cell responses to acute viral infection. BM chimeras were established and treated with tamoxifen as in *Figure 1C*, and infected with LCMV. The infected mice were analyzed on eight *dpi* for effector (A–D) and ≥40 *dpi* for memory phase responses (E, F). (A) Detection of CXCR5+SLAM$^{lo}$ T$_{FH}$ and CXCR5−SLAM$^{hi}$ T$_H$1 cells in CD45.2+ CD44$^{hi}$CD62L− activated CD4+ splenocytes on eight *dpi* by cell surface staining. (B) Detection of CXCR5+Bcl6+ GC-T$_{FH}$ cells in CD45.2+ CD44$^{hi}$CD62L− activated CD4+ splenocytes on eight *dpi* by intranuclear
*Figure 2 continued on next page*

Figure 2 continued

staining. (C) Detection of IFN-γ and/or TNF-α-producing cells in CD45.2$^+$CD8$^+$ splenocytes on eight *dpi* by intracellular staining after 5 hr incubation with GP33 peptides. (D) Detection of granzyme B expression in CD45.2$^+$ CD11a$^{hi}$ activated CD8$^+$ splenocytes on eight *dpi* by intracellular staining. Values in half-stacked histograms denote geometric mean fluorescence intensity (gMFI). (E) Detection of CXCR5$^+$ memory T$_{FH}$ cells in CD45.2$^+$CD44$^{hi}$ antigen-experienced CD4$^+$ splenocytes on ≥40 *dpi* by cell surface staining. (F) Detection of IFN-γ and/or TNF-α-producing memory CD8$^+$ T cells in CD45.2$^+$CD8$^+$ splenocytes on ≥40 *dpi* by intracellular staining after 5 hr incubation with GP33 peptides. In all panels, values in representative contour plots denote percentages, and cumulative data are means ± s.d. from two experiments. *, p<0.05 by Student's t-test; all other unmarked parameters were not statistically significant among the groups as determined by one-way ANOVA (A–D) or Student's t-test (E, F).

The online version of this article includes the following source data for figure 2:

**Source data 1.** Source files, containing original data for *Figure 2A–F*, to document frequency of antigen-specific Tfh cells (A, B) and CD8 T cells (C), expression of Granzyme B in CD8 T cells (D) at the effector response phase, and the frequency of antigen-specific Tfh cells (E) and CD8 T cells (F) at the memory phase.

frequency among all genotypes (*Figure 3B*), suggesting that loss of β/γ-catenin did not perturb HSC homeostasis.

Unlike HSCs, AML LSCs have been reported to critically depend on β-catenin using the MLL-AF9 or MLL-ENL mouse models (*Wang et al., 2010*; *Yeung et al., 2010*). To reappraise this requirement with our new animal model, we used the same MLL-AF9 AML model by retroviral delivery of the fusion protein (co-expressed with GFP) into Lin$^-$ BM cells from tamoxifen-treated CreER$^+$ WT or CreER$^+$Ctnnb1$^{fl/fl}$ BM chimeras (*Figure 3C*). The MLL-AF9-GFP-infected Lin$^-$ BM cells were then transplanted into CD45.1$^+$ primary (1°) recipients (*Figure 3C*). Because tamoxifen-mediated target gene ablation may not achieve 100% deletion efficiency in all cells, rare non-deleted cells could have growth advantage over β-catenin-deficient cells, especially in long-term studies such as LSC serial transplantation. To eliminate potential outgrowth of the rare non-deleted cells, we took the approach of recurring tamoxifen treatments at a standardized interval. Our optimizing experiments found that following the initial 4 doses of tamoxifen administration, 3 doses of tamoxifen delivery at 4 week intervals sustained elimination of β-catenin protein in *Ctnnb1*-targeted cells with minimal impact on hematopoietic cellularity. The efficacy of this regimen was validated by immunoblotting for β-catenin in CD45.2$^+$GFP$^+$Mac1$^+$ AML cells in the BM of 1° recipients (*Figure 3D*). The 1° recipients in both groups showed similar rate of survival (*Figure 3E*), suggesting that β-catenin null mutation did not affect the onset of AML.

To specifically investigate self-renewal of AML LSCs, we sort-purified CD45.2$^+$GFP$^+$Lin$^-$Mac1$^{lo}$c-Kit$^{hi}$ Sca1$^{lo}$CD16/CD32$^{hi}$ cells as AML LSCs from the 1° recipients (see *Figure 3—figure supplement 1* for gating strategy), followed by transplantation into secondary (2°) recipients (*Figure 3C*). Both WT and β-catenin-deficient AML LSCs showed similar capacity of propagating AML as determined by longitudinal tracking of CD45.2$^+$Mac1$^+$ GFP$^+$ AML leukemia burden in peripheral blood cells (PBCs, *Figure 3F*), and by the survival rate of 2° recipients (*Figure 3E*). In an independent experiment, 2° recipients of β/γ-catenin-deficient AML LSCs showed similar survival as those of WT or β-catenin-deficient LSCs (*Figure 3—figure supplement 2*). These data collectively argue against an essential role of β-catenin for AML LSCs.

To further substantiate the unexpected lack of impact on AML LSCs by null mutation of β/γ-catenin, we extended our AML studies to determine a requirement for Tcf/Lef family TFs that interact with β-catenin. There are four members in the Tcf/Lef family: Tcf1, Lef1, Tcf3 and Tcf4 (encoded by *Tcf7*, *Lef1*, *Tcf7l1*, and *Tcf7l2*, respectively). To fully address functional redundancy among Tcf/Lef family TFs, we used *Tcf7l1*-targeted murine embryonic stem (ES) cells generated by the International Knockout Mouse Consortium (*Figure 4A*) to produce *Tcf7l1*$^{fl/fl}$ mouse strain. By crossing with CreER and *Tcf7l2*$^{fl/fl}$ (*Angus-Hill et al., 2011*) strains, we generated CreER$^+$*Tcf7l1*$^{fl/fl}$*Tcf7l2*$^{fl/fl}$ (CreER$^+$Tcf3/4-dKO) mice. This line was further crossed with our established CreER$^+$*Tcf7*$^{fl/fl}$*Lef1*$^{fl/fl}$ (CreER$^+$Tcf1/Lef1-dKO) mice (*Yu et al., 2016*) to generate CreER$^+$*Tcf7*$^{fl/fl}$*Lef1*$^{fl/fl}$*Tcf7l1*$^{fl/fl}$*Tcf7l2*$^{fl/fl}$ mice (called CreER$^+$Tcf-qKO herein). Following treatment with the same tamoxifen regimen as above, the CreER$^+$Tcf-qKO mice did not exhibit early lethality, and Tcf1 and Lef1 proteins were ablated in thymocytes of CreER$^+$Tcf1/Lef1-dKO mice as determined by intracellular staining (*Figure 4B*). While ablation of Tcf4 protein was also effective in thymocytes as determined by immunoblotting (*Figure 4B*), Tcf3 protein was not reliably detected in thymocytes or BM cells (not shown), likely due to its expression in very low abundance (see *Figure 4C*). It should be noted that deletion efficacy for

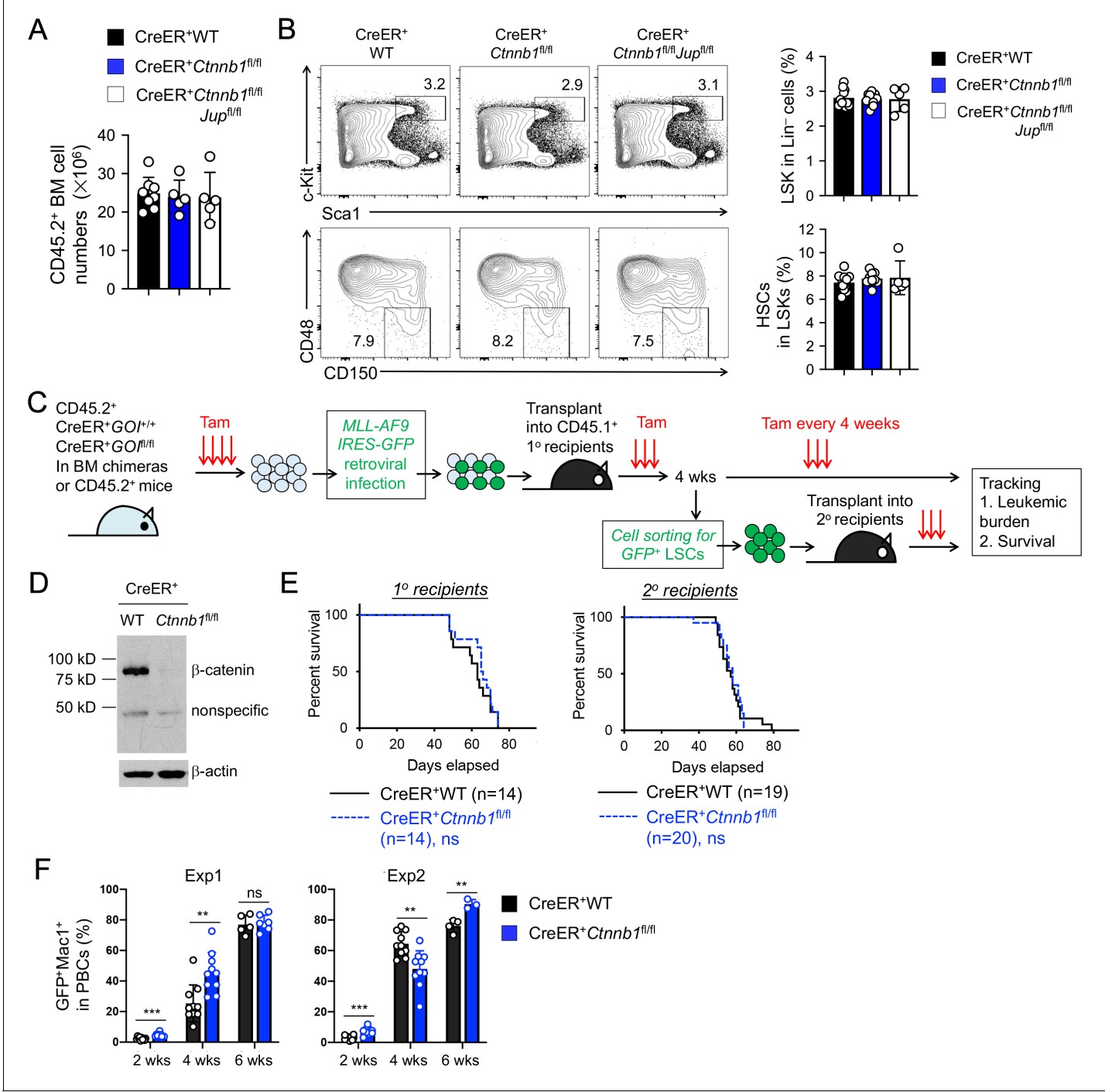

**Figure 3.** β-catenin is not essential for HSC homeostasis and AML LSC self-renewal. (A–B) BM chimeras were established and treated with tamoxifen as in *Figure 1C*, and analyzed for CD45.2+ BM cellularity (A), LSKs and HSCs by cell surface staining (B). Values in representative contour plots (B) denote percentages, and cumulative data on frequency of LSKs or HSCs are means ± s.d. from two experiments. None of the parameters was statistically significant among the groups as determined by one-way ANOVA. (C) Experimental design for modeling AML initiation and propagation in mice. After initial tamoxifen (Tam) treatments for four consecutive days, the 1° recipients and 2° recipients were subjected to recurring Tam treatment for three consecutive days at 4 week intervals to ensure long-term elimination of targeted proteins from the floxed alleles. For 1° recipients, MLL-AF9 retrovirus-infected Lin−BM cells containing $10^4$ GFP+Lin−cKit+ cells were transplanted. For 2° recipients, $10^3$ LSCs were sorted from 1° recipients at 4 weeks after initial transplantation and then transplanted. GOI, gene of interest. (D) Complete deletion of β-catenin in AML cells as determined by immunoblotting of sorted CD45.2+GFP+Mac1+ BM cells from 1° recipients on day 28 after BM transplantation, where WT and *Ctnnb1*-floxed cells were subjected to two rounds of tamoxifen treatment. (E) Kaplan-Meier survival curves of 1° and 2° recipients of WT or β-catenin-deficient LSCs. Data are pooled from two

*Figure 3 continued on next page*

**Figure 3 continued**

independent experiments. ns, not statistically significant as determined by log-rank test. (F) Longitudinal tracking of CD45.2$^+$GFP$^+$Mac1$^+$ AML leukemic cells in PBCs of 2° recipients. For week 8, the surviving recipients were analyzed. Data from two independent experiments were displayed separately because modest differences were observed in kinetics of leukemic cell expansion at week 4. These differences did not affect recipient survival (see **E**). **, p<0.01; ***, p<0.001 by Student's t-test.

The online version of this article includes the following source data and figure supplement(s) for figure 3:

**Source data 1.** Source files, containing original data for *Figure 3A,B and F*, to document BM cellularity (**A**), and frequency of LSKs and Slam HSCs in BM cells (**B**), and leukemia burden in peripheral blood of AML receipt mice (**F**).

**Figure supplement 1.** Gating strategy for sorting of AML LSCs, as defined as CD45.2$^+$GFP$^+$Lin$^-$Mac1$^{lo}$c-Kit$^{hi}$ Sca1$^{lo}$CD16/CD32$^{hi}$ BM cells in the 1° recipients on day 28 after BM transplantation.

**Figure supplement 2.** Kaplan-Meier survival curve of 2° recipients of WT, β-catenin or β/γ-catenin-deficient AML LSCs. ns, not statistically significant as determined by log-rank test.

Tcf3 protein was independently verified in mouse embryonic fibroblasts in a recent report using the same ES clone (*Mašek et al., 2016*). We further verified with quantitative RT-PCR that the targeted exon in *Tcf7l1* transcripts was as effectively deleted as that in *Tcf7l2* transcripts, in spite of the low abundance (*Figure 4C*). Tamoxifen treatment resulted in >90% reduction in thymic cellularity in CreER$^+$Tcf1/Lef1-dKO and CreER$^+$Tcf-qKO mice, but did not detectably affect CreER$^+$Tcf3/4-dKO thymocyte numbers (*Figure 4D*). Consistent with a requirement for Tcf1 for DP thymocyte survival (*Ioannidis et al., 2001*), tamoxifen-treated CreER$^+$Tcf1/Lef1-dKO and CreER$^+$Tcf-qKO mice showed profound reduction in DP thymocyte frequency, with concomitant relative increase in CD4$^+$ and CD8$^+$ single positive thymocyte frequency (*Figure 4E*). Within the DN compartment, CreER$^+$Tcf1/Lef1-dKO and CreER$^+$Tcf-qKO mice both showed DN1 accumulation and apparent loss of DN2 thymocytes (*Figure 4F*), in line with a key function of Tcf1 in specification of early thymic progenitors to T cell lineage (*Germar et al., 2011*; *Weber et al., 2011*). In all these critical aspects, CreER$^+$Tcf3/4-dKO mice exhibited little or modest changes; furthermore, the defects observed in CreER$^+$Tcf1/Lef1-dKO mice were not detectably more exacerbated in CreER$^+$Tcf-qKO mice with additional deletion of Tcf3 and Tcf4. Taken together, these observations suggest that Tcf3 and Tcf4 are not essential for thymopoiesis and further validate critical function of Tcf1 and Lef1 among the Tcf/Lef family in T cell development.

Given the demonstrated efficacy of ablating Tcf/Lef factors in hematopoietic cells, we extended our studies to investigate their requirement in AML LSCs. We used Lin$^-$ BM cells from tamoxifen-treated CreER$^+$ WT or CreER$^+$Tcf-qKO mice directly for the AML studies, without establishing BM chimeras. The 1° recipients of MLL-AF9-infected Tcf-qKO BM cells showed similar survival rate as those of WT BM cells (*Figure 4G*), indicating ablation of all four Tcf/Lef TFs did not affect AML onset. In addition, WT and Tcf-qKO AML LSCs isolated from the 1° recipients showed similar capacity of propagating AML leukemic cells in PBCs in 2° recipients (*Figure 4H*), resulting in similar survival rate of the 2° recipients (*Figure 4G*). Collectively, our data do not support an essential requirement for Tcf/Lef and β-catenin proteins in AML LSCs. It is of note that the lack of impact on AML LSCs by ablating Tcf/Lef TFs or β-catenin was not due to technical issues, because we have used the same protocol to demonstrate that 1) constitutive activation of non-canonical NF-κB pathway (*Xiu et al., 2018*) or 2) genetic ablation of Groucho/TLE corepressors (to be described elsewhere) impairs AML LSC self-renewal.

By use of the newly established β-catenin null mutation, this study clarifies on a long-standing controversy and provides more conclusive evidence indicating that β-catenin and γ-catenin are not essential for thymocyte development, T cell responses to viral infection at either effector or memory phase. This is in stark contrast to the versatile functions described for Tcf1 and Lef1 in T lineage cells (*Raghu et al., 2019*). We previously showed that deletion of the N-terminal domain in Tcf1, which abrogates interaction with β-catenin, does reduce thymic cellularity and compromise T$_{FH}$ cell differentiation at effector and memory phases (*Gullicksrud et al., 2017*; *Xu et al., 2017*). Given the apparent lack of impact upon β/γ-catenin deletion, Tcf1 may engage coactivators other than β-catenin to exert those biological functions. Indeed, we recently reported that Tcf1 interacts with a Ser21-phosphorylated form of Ezh2 to cooperatively induce Bcl6 and Icos during T$_{FH}$ cell differentiation in response to acute viral infection (*Li et al., 2018*). It may thus be fruitful to search for novel Tcf1 partners in developing thymocytes and antigen-responding mature T cells (*Steinke and Xue, 2014*).

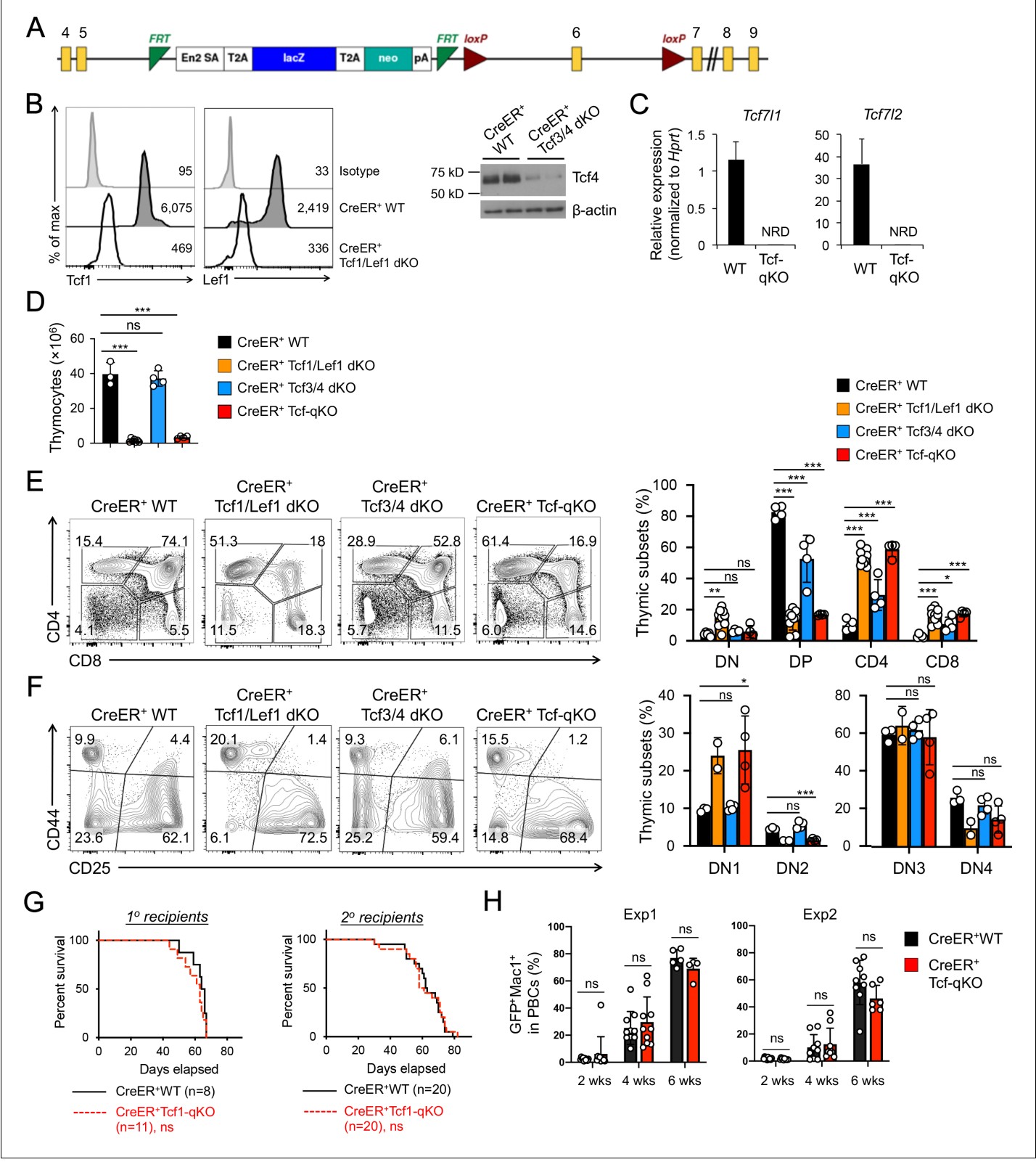

**Figure 4.** Tcf/Lef TFs are critical for T cell development but not essential for AML LSC self-renewal. (**A**) Targeting strategy for *Tcf7l1* gene locus. Yellow boxes denote exons, with exon numbers marked on top. Exon six is flanked with two LoxP sites (marked with red triangles). The LacZ/Neo cassette flanked by Frt Sites (cyan wedges) was removed with Flippase in germline-transmitted mice. (**B**) In vivo ablation of Tcf/Lef proteins. CreER+ WT, Tcf1/Lef1 dKO, and Tcf3/4 dKO mice were treated with tamoxifen for four consecutive days. Three days later, total thymocytes were intracellularly stained for

*Figure 4 continued on next page*

*Figure 4 continued*

Tcf1 and Lef1 in Tcf1/Lef1 dKO and control mice, and values denote geometric mean fluorescent intensity in representative half-stacked histograms (left panels). Total thymocytes were immunoblotted for Tcf4 protein in Tcf3/4 and control mice (right panel). (**C**) Validation of efficient deletion of targeted *Tcf7l1* and *Tcf7l2* exons in hematopoietic stem/progenitor cells. CreER⁺Tcf-qKO and WT mice were treated with tamoxifen for four consecutive days, and two days later, Flt3⁻LSK cells, which were enriched in both long-term and short-term HSCs, were sort-purified and analyzed by quantitative RT-PCR. Relative expression of *Tcf7l1* and *Tcf7l2* was determined by normalizing to *Hprt*, and shown as means ± s.d. (n = 5). NRD, not reliably detected. (**D**) Thymic cellularity. CreER⁺ WT, Tcf1/Lef1 dKO, Tcf3/4 dKO, and Tcf-qKO mice were treated with tamoxifen as in B), and thymocytes were enumerated. (**E**) Detection of thymic maturation stages. Thymocytes were surface-stained with biotinylated lineage markers (minus CD3ε) to exclude non-T cells, and with CD4 and CD8 to identify DN, DP, CD4⁺ and CD8⁺ subsets. (**F**) Detection of DN subsets. DN thymocytes were surface-stained with CD44 and CD25 to identify DN1 to DN4 subsets. Note that deletion of Tcf1 and Lef1, as in Tcf1/Lef1 dKO and Tcf-qKO mice, caused premature, modest upregulation of CD25 in a portion of DN1 cells, and the gate was adjusted accordingly to demarcate DN1 and DN2 subsets. Data in D–F are means ±s.d. from ≥2 experiments. Statistical significance for multiple groups was first assessed by one-way ANOVA, and that for indicated pair comparison was determined with Tukey's correction. *, p<0.05; **, p<0.01; ***, p<0.001; ns, not statistically significant. In F), n = 2 for Tcf1/Lef1 dKO, and thus no p values are shown. (**G**) Kaplan-Meier survival curves of 1° and 2° recipients of WT or Tcf-qKO LSCs. Data are pooled from two independent experiments. ns, not statistically significant as determined by log-rank test. (**H**) Longitudinal tracking of CD45.2⁺GFP⁺Mac1⁺ AML leukemic cells in PBCs of 2° recipients. For week 8, the surviving recipients were analyzed. Data from two independent experiments were displayed separately because modest differences were observed in kinetics of leukemic cell expansion at week 4. These differences did not affect recipient survival (see **G**). ns, not statistically significant as determined by Student's *t*-test.

The online version of this article includes the following source data for figure 4:

**Source data 1.** Source files, containing original data for *Figure 4D–H*, to document thymic cellularity (**D**), frequency of thymocyte subsets at different developmental stages (**E, F**), and leukemia burden in peripheral blood of AML receipt mice (**G**).

It is surprising that ablation of neither β-catenin nor Tcf/Lef TFs affected LSC self-renewal in the AML mouse model. The reported requirements for β-catenin using previously *Ctnnb1*-targeted mouse strains may be likely ascribed to unwanted effects by the truncated β-catenin protein. Although genetic alterations in Wnt-β-catenin pathway components are not among the primary driver mutations of AML (*Papaemmanuil et al., 2016*; *Tyner et al., 2018*), frequent translocation products including MLL-AF9 result in elevated β-catenin accumulation (*Lane et al., 2011*; *Müller-Tidow et al., 2004*). Our data do not refute the facts that aberrant activation of β-catenin has pathological effects, and there is no denial of therapeutic values in inhibiting Wnt-β-catenin pathway for debulking AML blasts. The findings in this study nonetheless caution that for the purpose of eradicating AML LSCs, targeting β-catenin may not be as effective as previously hoped.

# Materials and methods

## Key resources table

| Reagent type (species) or resource | Designation | Source or reference | Identifiers | Additional information |
|---|---|---|---|---|
| Genetic reagent (*Mus. musculus*) | C57BL/6J | Jackson Laboratory | RRID:IMSR_JAX:000664 | |
| Genetic reagent (*Mus. musculus*) | *Ctnnb1*ᶠˡ/ᶠˡ | This paper | | *Ctnnb1* exons 2–14 floxed. Send reagent request to haihui.xue@hmh-cdi.org |
| Genetic reagent (*Mus. musculus*) | *Jup1*ᶠˡ/ᶠˡ | PMID:22036570 | | |
| Genetic reagent (*Mus. musculus*) | *Tcf7*ᶠˡ/ᶠˡ | PMID:24836425 | | |
| Genetic reagent (*Mus. musculus*) | *Lef1*ᶠˡ/ᶠˡ | PMID:23103132 | | |
| Genetic reagent (*Mus. musculus*) | *Tcf7l1*ᶠˡ/ᶠˡ | This paper | | *Tcf7l1* Exon six floxed. Send reagent request to haihui.xue@hmh-cdi.org |

*Continued on next page*

*Continued*

| Reagent type (species) or resource | Designation | Source or reference | Identifiers | Additional information |
|---|---|---|---|---|
| Genetic reagent (*Mus. musculus*) | *Tcf7l2*<sup>fl/fl</sup> | PMID:21383188 | | |
| Strain, strain background (*virus*) | Lymphocytic choriomeningitis virus Armstrong strain E-350 | ATCC | VR-1271 | |
| Antibody | anti-β-catenin (Mouse monoclonal) | BD Biosciences | Cat. No. 610154 Clone 14 | IB (1:2000) |
| Antibody | anti-γ-catenin (Mouse monoclonal) | BD Biosciences | Cat. No. 610253 Clone 15 | IB (1:2000) |
| Antibody | Anti-Tcf4 (Rabbit monoclonal) | Cell Signaling Technology | Cat. No. 2565 Clone C9B9 | IB (1:1000) |
| Recombinant DNA reagent | MLL-AF9-GFP plasmid | PMID:17463288 | | |
| Chemical compound, drug | 4-hydroxy-tamoxifen | Millipore-Sigma | Cat. No. T176 | |
| Chemical compound, drug | Tamoxifen | Millipore-Sigma | Cat. No. T5648 | |
| Commercial assay or kit | Vybrant FAM caspase -3/7 assay kit | Invitrogen | Cat. No. V35118 | |
| Commercial assay or kit | QuantiTect Reverse Transcription Kit | Qiagen | Cat. No. 205313 | |
| Software, algorithm | FlowJo | https://www.flowjo.com | RRID:SCR_008520 | |
| Software, algorithm | GraphPad Prism | http://www.graphpad.com/ | SCR_015807 | |

## Animals and generation of BM chimeras

*Ctnnb1*<sup>fl/fl</sup> and *Tcf7l1*<sup>fl/fl</sup> mice were generated in this study. The following mouse strains were previously described, *Ctnnb1*-exons 2–6 floxed strain (Stock No. 004152, the Jackson Laboratory) (*Brault et al., 2001*), *Jup*<sup>fl/fl</sup> (*Demireva et al., 2011*), *Tcf7*<sup>fl/fl</sup> (*Steinke et al., 2014*), *Lef1*<sup>fl/fl</sup> (*Yu et al., 2012b*), *Tcf7l2*<sup>fl/fl</sup> (*Angus-Hill et al., 2011*), and *Rosa26*<sup>Cre-ERT2/+</sup> (*Guo et al., 2007a*). Transplantation of BM cells was performed as previously reported (*Li et al., 2017*; *Yu et al., 2012a*). All mouse experiments were performed under protocols approved by the Institutional Animal Use and Care Committee of the University of Iowa (Protocol No. 8021178) and Center for Discovery and Innovation, Hackensack University Medical Center (Protocol No. 276.00).

## Conditional targeting of the *Ctnnb1* locus

The *Ctnnb1*<sup>fl/fl</sup> mice were generated using Clustered Regularly Interspaced Short Palindromic Repeats (CRISPR) technology by Applied StemCell Inc. In brief, two LoxP cassettes were inserted into introns 1 and 14, flanking exons 2 to 14 of the *Ctnnb1* locus (*Figure 1A*). A mixture containing active guide RNA molecules (gRNA) and Cas-9 protein was injected into the cytoplasm of C57BL/6J (B6) embryo. The gRNA sequence for intron 1 is 5'- ACTGCTCTGACTTCACCCGAggg, that for intron 14 is 5'- CTATCATCACTCTATCCCAGagg. The pups born from the microinjection were screened by PCR and further confirmed by sequencing. Germline-transmitted F1 progeny was crossed with Cre-expressing strains for further analysis. For genotyping, the following primers were used to amplify genomic DNA: F1, 5'-CTGCTTACAGTGTGAGACACC; R1, 5'- CCAGTACTGCTC TGACTTCAC; and R2, 5'- CTGCCTGTCACAGATCAGATG. The combination of F1 and R1 detected *Ctnnb1* WT allele at 144 bp, and *Ctnnb1*-floxed allele at 178 bp, and the combination of F1 and R2 detected *Ctnnb1*-deleted allele at 218 bp. For detection of the *Ctnnb1* transcripts in RT-PCR, the following primers were used: Ex13F, 5'-GTCCTATTCCGAATGTCTGAGG; Ex15R, 5'-GGCCAGCTGA TTGCTATCAC.

### Verification of conditional deletion of the *Jup* locus

For detecting deletion of *Jup*-floxed exons on genomic DNA level, the following primers were used: F1, 5'- CTTCTGGGATCTCAGGAGTGTAC; R1, 5'- GTCATGTGCTAGCCCAGTCTAAG; and R2, 5'-TCACAGCCACTACCACTGAC. The combination of F1 and R1 detected *Jup*-floxed allele at 250 bp, and the combination of F1 and R2 detected *Jup*-deleted allele at approximately 280 bp. For detection of the *Jup* transcripts in RT-PCR, the following primers were used: Ex5F, 5'- AGACGGGCTGCAGAAGATG; Ex6R, 5'- GGGCTTGTTGCTAGGACAC.

### Conditional targeting of the *Tcf7l1* locus

The *Tcf7l1*-floxed embryonic stem (ES) cells were generated by the European Conditional Mouse Mutagenesis Program (EUCOMM), currently part of the International Knockout Mouse Consortium (IKMC) (*Figure 4A*). The allele is designated as *Tcf7l1*tm1a(EUCOMM)Wtsi (http://www.informatics.jax.org/allele/MGI:4432867). Microinjection of ES cells was performed at the Wellcome Trust Sanger Institute (UK), and germline-transmitted mice were bred with Flippase recombinase (expressed in the *Rosa26* locus, Jackson Laboratory, Stock No. 003946) to remove the LacZ/Neomycin cassette flanked by FRT sites. For genotyping of the resulting *Tcf7l1*fl/+ allele, the following primers were used to amplify genomic DNA: 5'-AGCAACCAAATGAAGGCTCAC and 5'-CTGCTGCCCCTCTTTTCATC, which detects *Tcf7l1* WT allele at 335 bp, and *Tcf7l1*-floxed allele at 424 bp, and *Tcf7l1*-deleted allele at 562 bp. Effective ablation of Tcf3 protein in *Tcf7l1*fl/fl mice was demonstrated by an independent study using the same ES cell-derived allele (*Mašek et al., 2016*). We also confirmed efficient CreER-mediated deletion of the floxed exon six in *Tcf7l1* transcripts in BM Flt3⁻LSKs by RT-PCR using the following primer sets: 5'-TCACCTACAGCAACGACCAC and 5'-TACGGTGACAGCTCAGATGG, with the latter complementary to the targeted exon 6 (*Figure 4C*). Deletion of the floxed exon one in *Tcf7l2* transcripts by CreER was confirmed in BM Flt3⁻LSKs by RT-PCR using 5'-ATGTCAAGTCCTCGCTGGTC and 5'-CCCTTAAAGAGCCCTCCATC primers, with the former complementary to the targeted exon 1 (*Figure 4C*).

### Induced target gene deletion by tamoxifen treatment

For ex vivo treatment, lineage-depleted BM cells from CreER⁺WT or CreER⁺*Ctnnb1*fl/fl mice were cultured in IMDM medium supplemented with 15% fetal bovine serum (FBS), 20 ng/ml thrombopoietin (TPO), and 50 ng/ml stem cell factor (SCF). Positively selected CD3⁺ T cells were cultured in RPMI-1640 medium supplemented with 10% FBS, 50 ng/ml IL-7, and 50 ng/ml IL-15. On second day of culture, 4-hydroxy-tamoxifen (T176, Millipore-Sigma) was added to a final concentration of 250 nM, and cells were harvested 3–5 days later for immunoblotting. For in vivo treatment, the mice were administered with tamoxifen (T5648, Millipore-Sigma) at 0.2 mg/g body weight via oral gavage for four consecutive days, as illustrated in *Figure 1C*. For long-term studies including serial transplantation of AML LSCs, the mice were given 3-day treatment every 4 weeks till the experimental endpoints, as illustrated in *Figure 3C*.

### Immunoblotting

Cell lysates were prepared from sorted CD45.2⁺ thymocytes, splenic T cells, BM cells, or GFP⁺-Mac1⁺ AML cells, resolved on SDS-PAGE, followed by immunoblotting with anti-β-catenin (Clone 14/Beta-Catenin, Cat. No. 610154, mouse monoclonal, BD Transduction Laboratories), with anti-β-actin (clone I-19, Santa Cruz Biotechnology) detection as control for equal loading. Thymocytes, BM and liver cells were detected with anti-γ-catenin (Cat. No. 610253, BD Biosciences) or anti-Tcf4 (C9B9, Cell Signaling Technologies).

### Flow cytometry and cell sorting

Single cell suspension was prepared from thymus, spleen, and BM and surface-stained as previously described (*Li et al., 2018*; *Shao et al., 2019*). All fluorochrome-conjugated antibodies were from eBiosciences/ThermoFisher Scientific. The antibodies and their clone numbers are TCRβ (H57-597), CD4 (RM4-5), CD8α (53–6.7), CD25 (PC61.5), CD44 (IM7), anti-IFN-γ (XMG1.2), anti-TNFα (MP6-XT22), anti-Icos (C398.4A), CD45.2 (104), Sca1(D7), c-Kit (2B8), CD48 (HM48.1), CD16/CD32 (93), Mac1 (M1/70), CD62L (MEL-14), and Streptavidin (eBiosciences Cat. No. 48-4317-82). For lineage markers, the following biotinylated antibodies were used: B220 (RA3-6B2), CD3ε (145–2 C11), γδTCR

(GL3), NK1.1 (PK136), CD11c (N418), Gr-1 (RB6-8C5), Mac1 (M1/70), and TER-119 (TER-119). CXCR5 (L138D7) and SLAM (TC15-12F12.2) are from BioLegend and used in single step staining for $T_{FH}$ cells. For detection of Bcl6 (K112-91) and Ki67 (B56, both from BD Biosciences), Tcf1 (C63D9) and Lef1 (C12A5) and corresponding isotype control (rabbit mAb IgG DA1E, all from Cell Signaling Technologies), surface-stained cells were fixed and permeabilized with the Foxp3/Transcription Factor Staining Buffer Set (eBiosciences/ThermoFisher Scientific), followed by incubation with corresponding fluorochrome-conjugated antibodies. Active Caspsase-3/7 was detected using the Vybrant FAM caspase-3/7 assay kit (Invitrogen/Life Technologies) as described (*Li et al., 2018*). Peptide-stimulated cytokine production and detection by intracellular staining were as described (*Zhao et al., 2010*). Anti-human granzyme B (FGB12) and corresponding isotype control were from Invitrogen/Thermo-Fisher Scientific. Data were collected on FACSVerse (BD Biosciences) and analyzed with FlowJo software (Version X, TreeStar). For cell sorting, surface-stained cells were sorted on BD FACSAria II or FACSAria Fusion cell sorter.

## Viral infection

BM chimeric mice were infected intraperitoneally (*i.p*) with $2 \times 10^5$ PFU of LCMV-Armstrong strain to elicit acute viral infection (*Li et al., 2018*; *Shao et al., 2019*). Splenocytes were harvested from the infected mice at eight *dpi* or $\geq$40 *dpi* to characterize mature $CD4^+/CD8^+$ T cell responses at the effector and memory phases, respectively.

## AML mouse model and LSC secondary transplantation

The AML mouse model was established following similar protocol in our previous CML/AML studies (*Li et al., 2017*; *Xiu et al., 2018*; *Yu et al., 2012a*). In brief, $Lin^-$ BM cells (either from CreER$^+$-Ctnnb1$^{fl/fl}$ BM chimeras or directly from CreER$^+$Tcf-qKO mice, all treated with tamoxifen) were infected with MLL-AF9-GFP retrovirus. The infected cells containing $1 \times 10^4$ GFP$^+$c-Kit$^+$ cells were transplanted into lethally irradiated congenic mice along with $2 \times 10^5$ CD45.1$^+$ protector BM cells. The primary recipients were treated with tamoxifen during days 25–27 after BM transplantation, and one day later, the BM cells were harvested and sorted for CD45.2$^+$ AML LSCs following the gating strategy outlined in *Figure 3—figure supplement 1*. One thousand of AML LSCs were mixed with $2 \times 10^5$ CD45.1$^+$ protector BM cells and transplanted into secondary recipients, followed by tracking CD45.2$^+$GFP$^+$Mac1$^+$ AML leukemia burden in PBCs. Both primary and secondary recipients were subjected to repeated rounds of tamoxifen treatment for three consecutive days by the end of every 4 weeks, and monitored for survival.

## Statistical analysis

For comparison between two experimental groups, Student's *t*-test was used, with a two-tailed distribution assuming equal sample variance. For multiple group comparisons, one-way ANOVA was used to determine whether any of the differences between the means are statistically significant. As post hoc correction, Tukey's test was used to determine statistically significance between two groups of interest. Comparison of AML mouse survival among different experimental groups was determined by log-rank test using Prism8 software. Statistical parameters, including numbers of samples or recipient mice analyzed (n), descriptive statistics (means and standard deviation) are reported in the figures and figure legends. P values equal to or more than 0.05 are considered not statistically significant (marked as 'ns' or unmarked for clarity). P values of less than 0.05 are considered statistically significant, the following asterisk marks are used to indicate the level of significance: *, p<0.05; **, p<0.01; ***, p<0.001.

## Acknowledgements

We thank Drs. Niccolo Zampieri (Max Delbrück Center for Molecular Medicine in the Helmholtz Association, Berlin, Germany) and Thomas Jessell (formerly Columbia University) for sharing the Jup$^{fl/fl}$ mice, Dr. Yiying Zhang (the New York Obesity Nutrition Research Center, with funding support from the NIH, P30DK26687) for sharing Rosa26$^{Cre-ERT2/+}$ mice, Dr. Melinda L Angus-Hill (University of Utah) for sharing Tcf7l2$^{fl/fl}$ mice, and Dr. John Dick (University of Toronto) for providing the MLL-AF9-GFP retroviral construct. We thank the University of Iowa Flow Cytometry Core facility (J Fishbaugh, H Vignes and G Rasmussen) for cell sorting, and Radiation Core facility (A Kalen) for

mouse irradiation. This study is supported in-part by grants from the NIH (AI112579, AI121080 and AI139874) and the Veteran Affairs BLR and D Merit Review Program (BX002903) to H-HX.

## Additional information

### Funding

| Funder | Grant reference number | Author |
|---|---|---|
| National Institute of Allergy and Infectious Diseases | AI121080 | Hai-Hui Xue |
| National Institute of Allergy and Infectious Diseases | AI139874 | Hai-Hui Xue |
| U.S. Department of Veterans Affairs | BX002903 | Hai-Hui Xue |
| National Institute of Allergy and Infectious Diseases | AI112579 | Hai-Hui Xue |

The funders had no role in study design, data collection and interpretation, or the decision to submit the work for publication.

### Author contributions

Xin Zhao, Conceptualization, Data curation, Formal analysis, Supervision, Validation, Investigation, Writing - original draft, Writing - review and editing; Peng Shao, Data curation, Formal analysis, Validation, Investigation; Kexin Gai, Fengyin Li, Qiang Shan, Data curation, Formal analysis, Investigation; Hai-Hui Xue, Conceptualization, Supervision, Funding acquisition, Writing - original draft, Project administration, Writing - review and editing

### Author ORCIDs

Hai-Hui Xue (iD) https://orcid.org/0000-0002-9163-7669

### Ethics

Animal experimentation: All mouse experiments were performed under protocols approved by the Institutional Animal Use and Care Committees of the University of Iowa (Protocol No. 8021178) and Center for Discovery and Innovation, Hackensack University Medical Center (Protocol No. 276.00).

### Decision letter and Author response

Decision letter https://doi.org/10.7554/eLife.55360.sa1
Author response https://doi.org/10.7554/eLife.55360.sa2

## Additional files

### Supplementary files

• Transparent reporting form

### Data availability

Source data files provided. Mouse strain will be made available to other investigators on request.

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
