## [Decision Letter]

**Acceptance summary:**

The findings reported here add to the body of literature, often conflicting, on the role of β-catenin in particular in AML stem cells and T lymphocytes. In particular, the finding that the complete knockout of β-catenin and γ-catenin does not impair AML in primary or secondary recipients is an interesting and important finding with scientific and translational implications.

**Decision letter after peer review:**

Thank you for submitting your article "β-catenin and γ-catenin are dispensable for T lymphocytes and AML leukemic stem cells" for consideration by *eLife*. Your article has been reviewed by two peer reviewers, one of whom is a member of our Board of Reviewing Editors, and the evaluation has been overseen by Utpal Banerjee as the Senior Editor. The reviewers have opted to remain anonymous.

The reviewers have discussed the reviews with one another and the Reviewing Editor has drafted this decision to help you prepare a revised submission.

Summary:

In the manuscript by Zhao et al., the authors investigate the requirement for β- and γ-catenin in T cell development, T cell activation, and AML LSCs using a novel β-catenin knockout mouse that eliminates most exons of this gene. Prior studies investigating β-catenin in these processes used mice with limited exon targeting, raising the possibility that these mice actually expressed a truncated protein with potentially novel functions. Using the new mice alone or bred with γ-catenin knockout, the authors show that these proteins are not required for normal T cell development, T cell activation and viral immune responses, and AML development and LSC transplantation using the MLL-AF9 retroviral model. Moreover, deletion of all 4 LEF/TCF transcription factors also had no effect on AML development or LSC transplantation in this same model. In general, the role and requirement of β-catenin in T cell development and AML is of interest – and has been controversial due to confounding reports in the literature. However, there are several issues with the current manuscript.

Essential revisions:

1) One of the key issues with this manuscript is the discrepancy between the findings here and the results from the prior published studies with the alternative models. Thus, it is key to identify mechanisms that account for the discrepancy. Is there any evidence that the prior β-catenin mice targeted in exons 2/3-6 produce a truncated protein? Independent of these mice, can a truncated β-catenin be generated from the non-targeted exons? what is the phenotype of this truncated protein in relevant assays of β-catenin function?

2) In the Western blot in Figure 1D showing absence of β-catenin protein, a band is detected at 50kD labeled as non-specific. However, this band seems to be decreased in the β-catenin KO cells as well, raising the possibility that it is not non-specific. Can the authors comment on the identity of this band? Is it possibly an aberrant form of β-catenin?

3) The KO of γ-catenin should be validated with Western blot and DNA/RNA analysis as was done for β-catenin.

4) In Figure 4A,B, the authors conclude that normal hematopoiesis and HSCs are unaffected by β- and γ-catenin double KO. However, in order to formally make this conclusion, transplantation assays should be conducted showing no difference compared to WT control.

5) What is the efficacy of β- and γ-catenin knockout at time of primary mice analysis and secondary mice analysis as determined by Western blot? It is possible that rare non-deleted cells are growing out, particularly at later time points.

6) For the Tcf-qKO cells, Western blot should be shown demonstrating elimination of each protein. What is the effect of Tcf-qKO on T cell development and activation in the experiments shown in Figures 2,3?

7) Do the β-catenin and γ-catenin individual or double knockout mice develop T cell lymphoma?

---

## [Author Response]

Essential revisions:1) One of the key issues with this manuscript is the discrepancy between the findings here and the results from the prior published studies with the alternative models. Thus, it is key to identify mechanisms that account for the discrepancy. Is there any evidence that the prior β-catenin mice targeted in exons 2/3-6 produce a truncated protein? Independent of these mice, can a truncated β-catenin be generated from the non-targeted exons? what is the phenotype of this truncated protein in relevant assays of β-catenin function?

These are important points. Production of a truncated β-catenin protein from existing mouse models was previously reported by Held and colleagues (Jeannet et al., 2008). The authors examined bone marrow cells from both mouse models, exons 2-6 floxed model (the RK model, developed by R Kelmer and colleagues, Brault et al., 2001), and exons 3-6 floxed model (the JH model, developed by Birchmeier W and colleagues, Huelsken et al., 2000).

**Author response image 1. sa2fig1:** 

As shown in Author response image 1 panel A, single allele deletion of β-catenin in either model resulted in enhanced production of the truncated protein (lanes 2 and 3 for the RK and JH models, respectively). The truncated protein can be detected in non-targeted wild-type bone marrow cells in low amount (lane 1), which becomes more discernible after longer exposure (middle panel). It is of interest to mention that the truncated form of β-catenin was not detected in keratinocytes (panel B), suggesting a tissue-specific effect.In our own experiments, we previously (about 5 years ago) obtained the RK model (Stock No. 004152, the Jackson Laboratory) and crossed with Vav-Cre to ablate β-catenin. By immunoblotting bone marrow cell lysates with the same β-catenin antibody as above (clone 14, BD Transduction Laboratory), the truncated β-catenin was detected at approximately 40 kDa in the targeted cells, validating the findings by Held and colleagues. We now include the data in Figure 1—figure supplement 1, described the relevant information in the Results and Discussion to highlight this critical point.

As for the function of truncated forms of β-catenin protein, because the N-terminus of β-catenin contains phosphorylation sites for ubiquitin-dependent degradation, an N-terminal truncated form of β-catenin protein has been shown to have longer half-life, and ectopic expression of the N-terminal truncated β-catenin stimulates proliferation and apoptosis of intestinal crypts (Wong et al., 1998). In planarians (flatworms), a C-terminal truncated β-catenin is a naturally occurring paralog and acts as a negative regulator of full-length β-catenin during planarian eye photoreceptor specification (Su et al., 2017). We now cited these papers and described relevant findings in the Introduction.

2) In the Western blot in Figure 1D showing absence of β-catenin protein, a band is detected at 50kD labeled as non-specific. However, this band seems to be decreased in the β-catenin KO cells as well, raising the possibility that it is not non-specific. Can the authors comment on the identity of this band? Is it possibly an aberrant form of β-catenin?

As described in response to Point #1 and shown in Figure 1—figure supplement 1, the truncated β-catenin resulting from exon 7 translation in the RK model was detected at ~40 kDa. The non-specific band was at ~50 kDa. To substantiate our interpretation that the 50 kDa band results from non-specific reaction to the β-catenin antibody, we added the following two sets of data:

First, we treat lineage-negative bone marrow and peripheral T cells with tamoxifen in vitro. As shown in immunoblots in Figure 1B, the lineage-negative BM cells were completely devoid of non-specific bands; while T cells had the non-specific band, the signal intensity was similar between DMSO- and tamoxifen-treated cells. This was included as new Figure 1B, and described in the Results and Discussion first paragraph.

Second, we updated original Figure 1D which showed one set of data with the new Figure 1E which now includes two sets of data. The non-specific bands were not always detected (Thymocytes Exp1 and BM cells Exp2), and if detected, the non-specific bands did not always show decreased intensity in β-catenin KO cells (for example, BM cells Exp1, the non-specific band was stronger in KO cells, if any).

Taking the new Ctnnb1-targeting strategy into consideration, where Ctnnb1 exons 2-14 are floxed, the extensive immunoblot data suggest that the 50 kb band is unlikely a truncated form of β-catenin.

3) The KO of γ-catenin should be validated with Western blot and DNA/RNA analysis as was done for β-catenin.

As requested, we have designed primers based on the published γ-catenin (gene name, Jup) targeting strategy as summarized below and in Figure 1—figure supplement 4A.

On the genomic DNA level, we validated effective excision of the floxed Jup exons using the F1-R1 and F1-R2 primer combinations (Figure 1—figure supplement 4B).

On the mRNA level, we found that Jup transcripts were much less abundant compared with β-catenin transcripts in WT BM cells (only 1/50th of the latter after normalized to the Hprt housekeeping gene). Nonetheless, we were able to observe consistent deletion of Jup transcripts upon tamoxifen treatment in the KO bone marrow chimeras, after zooming in to Jup transcripts only (right panel in Figure 1—figure supplement 4C).

Because of the low abundance of Jup transcripts, the γ-catenin protein was below detection limit by immunoblotting in bone marrow cells (left panel Figure 1—figure supplement 4D). However, γ-catenin protein was more readily detectable in hepatocytes, and we confirmed induced deletion of γ-catenin by tamoxifen in non-BM chimera, KO mice (right panel Figure 1—figure supplement 4D).

Overall, these analyses validated effective ablation of γ-catenin in our experimental system.

4) In Figure 4A,B, the authors conclude that normal hematopoiesis and HSCs are unaffected by β- and γ-catenin double KO. However, in order to formally make this conclusion, transplantation assays should be conducted showing no difference compared to WT control.

We agree with the reviewers that if we were to conclude that ablation of β-/γ-catenin does not affect HSC function and self-renewal, it would be necessary to perform serial transplantation assays. Based on data in Figure 3A,B, we were careful to conclude that “loss of β-/γ-catenin did not perturb HSC homeostasis” but did not comment on HSC self-renewal. The rationale is that previous studies on HSCs using the existing RK or JH β-catenin targeted models did not observe a strong impact of deleting β-catenin on HSC self-renewal, even if truncated β-catenin protein may have been present in those analyses (Zhao et al., 2007 and Ruiz-Herguido et al., 2012). In contrast, previous studies on AML LSCs using these models reported a requirement for β-catenin for LSC self-renewal (Yeung et al., 2010 and Wang et al., 2010). We therefore focused on biological aspects where β-catenin was considered to have a prominent role, and this focus is reflected in manuscript title “T lymphocytes and AML leukemic stem cells”.

Nonetheless, we did perform one serial transplantation experiment on WT and CreER^+^*Ctnnb1*^fl/fl^ HSCs, where the contribution of WT and β-catenin deficient HSCs contributed similarly to blood reconstitution and LSK compartment in secondary recipients when analyzed at 6 weeks after transplantation (see Author response image 2). We are happy to include these data if required by the reviewers, but respectfully suggest that the manuscript would benefit from focusing on currently unsolved issues on β-catenin.

5) What is the efficacy of β- and γ-catenin knockout at time of primary mice analysis and secondary mice analysis as determined by Western blot? It is possible that rare non-deleted cells are growing out, particularly at later time points.

We thank the reviewers for raising this point, we are fully aware of the “escapee” cells that may have growth advantage over KO cells. We therefore have laid out an experiment plan (Figure 3C), where the tamoxifen was administered repeatedly to ensure that the rare non-deleted cells were eliminated for the duration of the entire study.

As depicted in Figure 3C in the AML studies, the BM chimeras were treated with tamoxifen for the initial induced deletion, which were then retrovirally transduced with MLL-AF9 and transplanted into the primary recipients. The primary recipients were treated with another round of tamoxifen on days 25, 26 and 27 after transplantation, so as to be sure that AML LSCs contained minimal, if any, non-deleted cells before transplanted into secondary recipients. Both primary and secondary recipients were continuously treated every 4 weeks.

It is a fine balance to weed out “non-deleted” cells and avoid unwanted side-effects resulting from too frequent tamoxifen treatments. The regimen of treatment at 4-weeks interval was optimized at the early stages of the studies. In fact, from the primary recipients, we sorted CD45.2^+^GFP^+^Mac1^+^ AML cells on day 28 after BM transplantation (i.e., after days 25-27 tamoxifen treatment), and validated absence of β-catenin protein in the KO AML cells (shown in Figure 3D). Although we did not perform the same experiment on AML cells from the secondary recipients, we deduce that the standardized tamoxifen regimen should have consistent effect in both primary and secondary recipients under the same experimental system.

We refined the description of the tamoxifen regimen and its purpose in the text and figure legends.

6) For the Tcf-qKO cells, Western blot should be shown demonstrating elimination of each protein. What is the effect of Tcf-qKO on T cell development and activation in the experiments shown in Figures 2,3?

As requested, we performed the following analyses:

1) To demonstrate efficiency of protein ablation, we used intracellular staining for Tcf1 and Lef1 in thymocytes, where both proteins are mostly abundantly expressed. As shown in Figure 4B, both Tcf1 and Lef1 proteins were greatly reduced to a level similar to isotype controls.

Immunoblotting also showed that Tcf4 protein was substantially reduced in thymocytes upon induced deletion. On the other hand, Tcf3 protein were not reliably detected in thymocytes and BM cells from WT or Tcf3/4 dKO mice. This is likely due to low abundance of Tcf7l1 expression in hematopoietic cells, as reflected in RT-PCR analysis shown in Figure 4C, where Tcf7l1 transcripts were about 1/25th of Tcf7l2 transcripts in Flt3–LSK cells. In spite of low expression, Tcf7l1 transcripts were ablated to an undetectable level in Tcf-qKO cells (Figure 4C). It is also important to note that the same Tcf7l1-targeted ES clone was used in another independent study to ablate Tcf3 in mouse embryonic fibroblasts (Masek et al., 2016), corroborating the efficacy of Tcf3 deletion using the same model. Because Tcf1, Lef1 and Tcf4 were deleted effectively in our experimental system, we respectfully suggest that it is reasonable to deduce similar capacity of Tcf3 protein in the same cells.

2) We analyzed T cell development in Tcf-qKO mice, along with CreER+ Tcf1/Lef1 KO and CreER+ Tcf3/Tcf4 KO mice. Tamoxifen treatment resulted in over 90% reduction in thymic cellularity in CreER+ Tcf1/Lef1 KO and Tcf-qKO mice, but did not detectably affect thymocyte numbers in CreER+ Tcf3/Tcf4 KO mice.

In CreER+ Tcf1/Lef1 KO and Tcf-qKO thymuses, the DP and DN2 thymocytes showed more pronounced reduction, with DN1 thymocytes showing premature CD25 upregulation and relative accumulation compared with CreER+ WT mice. These aspects are consistent with well-established requirements for Tcf1 in promoting T cell lineage commitment and DP thymocyte survival. In addition, none of these changes was evident in CreER+ Tcf3/Tcf4 KO mice, and the defects in CreER+ Tcf1/Lef1 KO mice were not further exacerbated in Tcf-qKO mice. These observations suggest that Tcf3 and Tcf4 are not required for thymopoiesis and further validate critical function of Tcf1 and Lef1 among the Tcf/Lef family. We now included all these data as Figure 4E and 4F, and described in the Results and Discussion.

The analysis of T cell development also validated the effectiveness of induced deletion of Tcf/Lef proteins in our experimental system, and thus lent additional support to the conclusion that Tcf/Lef factors and β-catenin are dispensable for AML LSC self-renewal. This was also the primary reason and purpose for inclusion of the Tcf-qKO model.

As for T cell activation in the Tcf-qKO model, we agree that is an important scientific question, but respectfully suggest it is beyond the focus on β/γ catenin in this manuscript. As shown in Figure 4E and F, induced deletion of Tcf/Lef proteins (primarily due to deletion of Tcf1) greatly diminished thymic output. As a result, the precursor frequency of antigen-specific T cells was reduced in the peripheral lymphoid organs when challenged with viral infection, making it a less ideal model for investigation of T cell activation and differentiation. This was not a major issue for induced deletion of β- and γ-catenin (Figure 2), because thymic development was not detectably affected. As such, analysis of the contribution of Tcf3/Tcf4 and their possible cooperativity with Tcf1/Lef1 should be performed using a model where these proteins are specifically deleted in mature T cells, like we did for Tcf1 and Lef1 in previous studies (Choi et al., 2015, and Shan et al., 2017). We hope that the reviewers/editors would allow us to address those questions in a separate, dedicated project.

7) Do the β-catenin and γ-catenin individual or double knockout mice develop T cell lymphoma?

We did monitor a cohort of WT and CreER^+^*Ctnnb1*^fl/fl^*Jup*^fl/fl^ BM chimeras for 16 weeks after tamoxifen treatment and did not observed signs of malignant transformation of thymocytes or other hematopoietic lineages. We included a statement on this observation.